# Effect of Lactamase Inhibitors on the Biosensor Penp during the Measurement of Lactam Antibiotics Concentration

**DOI:** 10.3390/s19051237

**Published:** 2019-03-12

**Authors:** Dagoberto Soto, Camila Silva, Cristian Ugalde, Kwok-Yin Wong, Yun-Chung Leung, Lok-Yan So, Max Andresen

**Affiliations:** 1Departamento de Medicina Intensiva, Facultad de Medicina y Hospital Clínico, Pontificia Universidad Católica de Chile, Santiago 8330024, Chile; dasotom@uc.cl (D.S.); cfsilvag2@gmail.com (C.S.); cristian.ugalde@gmail.com (C.U.); 2State Key Laboratory of Chemical Biology and Drug Discovery, Department of Applied Biology and Chemical Technology, The Hong Kong Polytechnic University, Hung Hom, Kowloon, Hong Kong; kwok-yin.wong@polyu.edu.hk (K.-Y.W.); thomas.yun-chung.leung@polyu.edu.hk (Y.-C.L.); annsoly@gmail.com (L.-Y.S.)

**Keywords:** biosensor, PenP, lactamase inhibitors, lactam antibiotics, use

## Abstract

PenP is a fluorescent biosensor of lactam antibiotics (LA). It is structurally derived from the mutant lactamase TEM-1 comprising the substitution E166C, where fluorescein is covalently linked to cysteine. The presence of LA in the medium produces a change in the intrinsic fluorescence level of the biosensor, and the integral of the fluorescence level over time correlates directly with the LA concentration. Previously, we have successfully used PenP to determine the concentration of lactam antibiotics in clinical samples. The use of lactamase inhibitors (LI) is a common strategy to enhance the effect of LA due to the inhibition of an important resistance mechanism of pathogenic microorganisms. Structurally, LI and LA share the common element of recognition of lactamases (the lactam ring), but they differ in the reversibility of the mechanism of interaction with said enzyme. Because the biological recognition domain of PenP is derived from a lactamase, LI is expected to interfere with the PenP detection capabilities. Surprisingly, this work provides evidence that the effect of LI is marginal in the determination of LA concentration mediated by PenP.

## 1. Introduction

A biosensor is a self-contained device capable of providing quantitative analytical information of a certain analyte, mediated by a biological recognition domain associated and a transduction element functionally integrated. In this work, PenP is a biosensor of lactam antibiotics (LA) wherein its biological recognition domain is a mutant lactamase derived from TEM-1, an enzyme that naturally exhibits affinity for and the ability to degrade LA. The mutation comprises the substitution of amino acid Glutamate by Cysteine, at the catalytic site in the position 166 (omega loops). The integrated fluorescent transduction element results from the covalent attachment of Fluorescein-5-maleimide to the cysteine residue at the Cysteine 166 [1]. The said substitution diminishes the catalytic activity of the mutant lactamase [2,3]. 

Previously, we developed a PenP-based method to determine LA concentration in a clinically relevant dynamic range (nM–mM). In fact, we have successfully used PenP to determine LA concentration in serum during complex clinical procedures [1,4,5]. 

The PenP detection mechanism is based on at least two steps:(1)First, the biological recognition domain of PenP interacts with LA and a dynamic equilibrium occurs in time that is a function of LA concentration.(2)Next, a change in the intrinsic fluorescence level of PenP occurs because a conformational change occurs at the biological recognition domain of PenP associated with LA.

The method based on PenP is quick and easy to implement, therefore it is desirable to expand the spectrum of substrates that can be measured with PenP [6]. 

The use of lactamase inhibitors (LI) is a common strategy to enhance the effect of LA because such agents inhibit a mechanism of resistance of microorganisms [7,8]. The LI has structural affinity for the lactamases because it comprises a lactam ring in its structure (same as LA) [9]. However, instead of a reversible interaction, the inhibition mediated by LI is based on “suicidal agents” that bind irreversibly to the active site of the enzyme (Table 1) [10], thus leading to its functional limitation.

As the PenP’s biological recognition domain is derived from a lactamase, it is expected that LI would interfere with the PenP sensing capacities, but this is still unknown. This work addresses the characterization of LI effects on PenP to determine the concentration of LA. 

## 2. Materials and Methods

### 2.1. Antibiotics 

Meropenem, cefazolin, benzylpenicillin, ampicillin, amoxicillin, amoxicillin/clavulanic acid and ampicillin/sulbactam, were acquired from Pharma investi and Laboratorios (Chile); clavulanic acid, sulbactam, BSA and all other chemical agents were acquired from Sigma-Aldrich; Biosensor: PenP E166C to a concentration of 5 × 10^−8^ M was prepared in PBS with 1% of BSA.

### 2.2. Fluorescence Kinetics

Multilector Sinergy was used to determine in 96-well plates the change of the fluorescence level over time of PenP induced by LA, LI or mixes thereof. The optic configuration was set up for fluorescein (excitation 485/20 nm; emission 528/20 nm). The fluorescence kinetics was recorded on each well with intervals of 60 s for 90 min. Each measurement was performed in triplicate for each of the compounds and/or compositions.

### 2.3. Treatment of the Data 

Data obtained from the multilector were analyzed as in Andresen et al. [6]. We previously observed that the pattern of fluorescence over time (kinetics of fluorescence) induced on PenP by different LA leads to a different pattern. For example, while meropenem exhibited a stable increase in fluorescence, cefazolin, amoxicillin, and ampicillin showed a transient increase in fluorescence (Figure 1a). Despite these differences, we previously demonstrated that the use of the area under the curve in a period of 90 min against the logarithm of the LA concentration exhibited a predictable Boltzmann or 4PL correlation.

### 2.4. Stats

The adjustments of the results to a Boltzmann curve were performed with GraphPad Prism V5. The results were expressed as the mean of the least three series with their respective standard deviation (mean ± SEM), calculated with GraphPad Prism.

## 3. Results

### 3.1. Fluorescence Pattern Induced on PenP Exposed to Different LA and Its Relation with the Lactamase Resistance

At all tested concentrations, Meropenem, a known LA resistant to the lytic action of lactamases, induced an increase in the fluorescence level of the PenP biosensor and was kept stable over time. In contrast, almost all tested concentrations of cefazolin, amoxicillin, or ampicillin induced a transient increase in the level of biosensor fluorescence that after reaching maximum rapidly returned to the baseline (Figure 1a). Consistent with our previous publication, the plurality of fluorescence patterns induced in the sensor by different antibiotics at different concentrations adjusted well to a Boltzmann curve (Figure 1b).

### 3.2. Fluorescence Pattern Induced on PenP Exposed to Different LI

Based on the above results, and considering the mechanism of interaction between the LI and lactamases, we worked on the hypothesis that the covalent linkage of LI to the biological recognition domain of PenP would induce a progressive increase over time in the fluorescence level of the biosensor, similar to the pattern observed for meropenem.

Clavulanate or sulbactam concentrations below 100 μM increased the fluorescence level of the biosensor, although far from the saturation level, at least during the first 90 min of reaction (Figure 2a). This indicates that a molar ratio of 2000 times of LI by each Biosensor was insufficient to deplete the functional mass of PenP. In the same range of concentrations, Clavulanate, as expected for its irreversibility, increased the fluorescence level of the biosensor in a stable manner. Surprisingly, sulbactam did in a transient manner. 

### 3.3. Fluorescence Pattern Induced on PenP Exposed to Different Commercial Compositions Comprising LA and LI

Previously, we demonstrated that the level of fluorescence change induced in the biosensor by the presence of LA is directly proportional to the biosensor mass arranged in the system [6]. Therefore, here we work in the hypothesis that the covalent linkage of LI to the biological recognition domain of PenP would result in a progressive loss of the functionally available mass of it to interact with LA, which would lead to at least an underestimation of the concentration of LA.

The changes in the fluorescence level of PenP induced by different concentrations of compositions comprising lactam antibiotics (at different concentrations; 10^−8^–10^−4^ M) and lactamase inhibitor (10^−6^ M), such as amoxicillin/clavulanate or ampicillin/sulbactam, were registered (Figure 3). Unexpectedly, the fluorescence pattern induced by these mixtures LA/LI over PenP was mainly dominated by the pattern induced by LA. 

## 4. Discussion

The transient quality of the fluorescence levels induced by antibiotics sensitive to the lactamase activity suggested that the biological recognition domain of PenP exhibited residual activity (Figure 1 in Cefazolin, Amoxicillin and Ampicillin). Therefore, this biosensor could be conveniently used to monitor the lactamase resistance of new lactam antibiotics in real time.

Given the excess molar ratio between LI and PenP and the interaction mechanism described between the LI and lactamases, we expected that the fluorescence level of LI-induced PenP would progressively increase over time until the functional mass of the available biosensor was depleted, this visualized as the increase in fluorescence up to the saturation level occurred. Surprisingly, at concentrations below 100 micromolar, the fluorescence pattern did not show the expected progressivity (Figure 2a). Since it is known that substitutions in the glutamic amino acid 166 did not interfere with the interaction between LI and lactamases [10], we assumed that the absence of progression in fluorescence is a particular phenomenon of PenP, possibly derived from reactivity limitations derived from steric impairment by the presence of the fluorescein.

The fluorescence level induced by LI at commercial concentrations on PenP was not successful in inducing a saturation level of the sensor (Figure 2a), consequently there is reason to state that in the media remains enough mass of the sensor to sense LA simultaneously.

Derived from our initial hypothesis, we expected that the presence of LI could interfere with the determination of LA concentration by PenP. Unexpectedly, the pattern of fluorescence induced in PenP mixtures by LI/LA was very similar to the pattern observed only with LA, showing a displacement of the Boltzmann curve to the left and resulting in an overestimation of 10–20% LA (Figure 3c).

Interestingly, the fluorescence pattern observed in LI was observed in the mixture but at a later stage, suggesting a slower reaction kinetics for said agents, consistent with the idea of a limited reactivity between LI and the biological recognition domain of PenP derived from steric deterioration due to the presence of fluorescein (Figure 3a).

## 5. Conclusions

In order to ensure the reproducibility of an assay dependent biosensor in general and PenP in particular, the biological recognition domain of an ideal biosensor based on enzymes should interact with its target in a reversible way in order to ensure to avoid changes of the functional mass available to sense the target. 

Considering that:(a)The mixtures of LI with LA are common because potentiate the therapeutic effects of the antibiotic,(b)The mechanism of interaction among LI/lactamase is irreversible, and(c)The biological recognition domain of PenP is a lactamase mutant.

There are reasons to suppose that LI could interfere in the analytical properties of PenP, therefore, it was necessary to determine the correction factors necessary for the previously developed method to be useful in the presence of such agents. Concentrations of LI below about 100 μM, such as those observed in commercial mixtures with antibiotics, induce an overestimation of LA of about 10–20%.

## Figures and Tables

**Figure 1 sensors-19-01237-f001:**
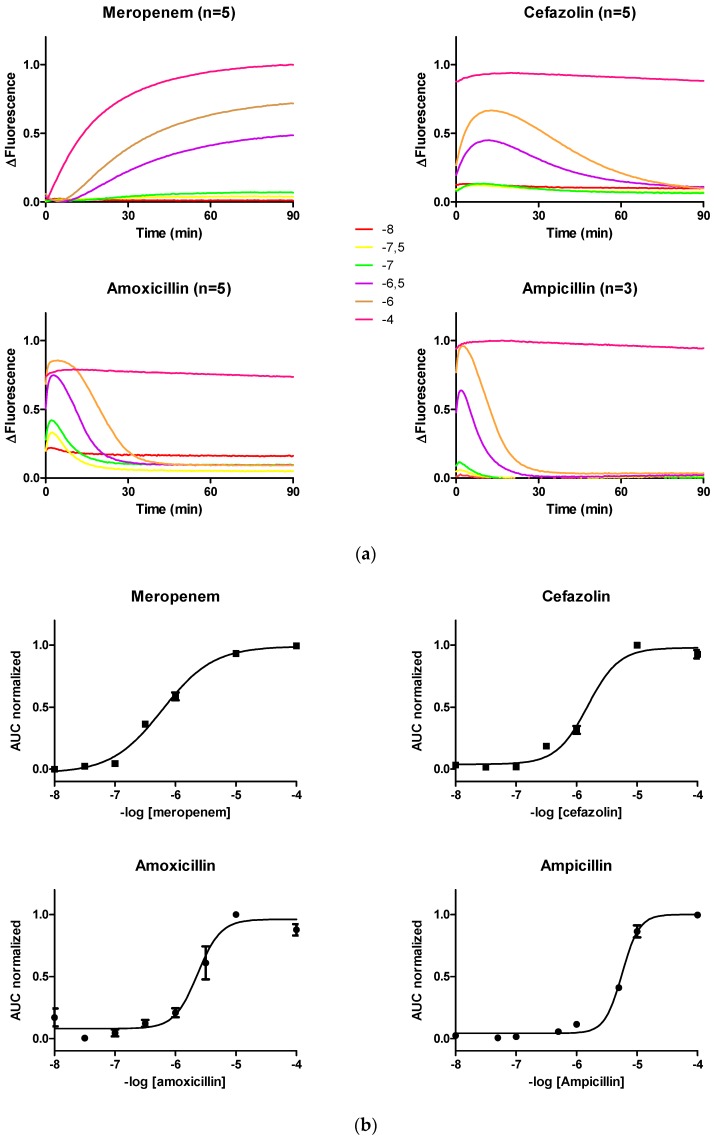
(**a**) The time course of the fluorescence level of PenP induced by different concentrations of meropenem, cefazolin, amoxicillin or ampicillin was recorded. (**b**) For each antibiotic, the area under the normalized fluorescence curve versus the logarithmic concentration of the antibiotic was adjusted to a Boltmzmann curve.

**Figure 2 sensors-19-01237-f002:**
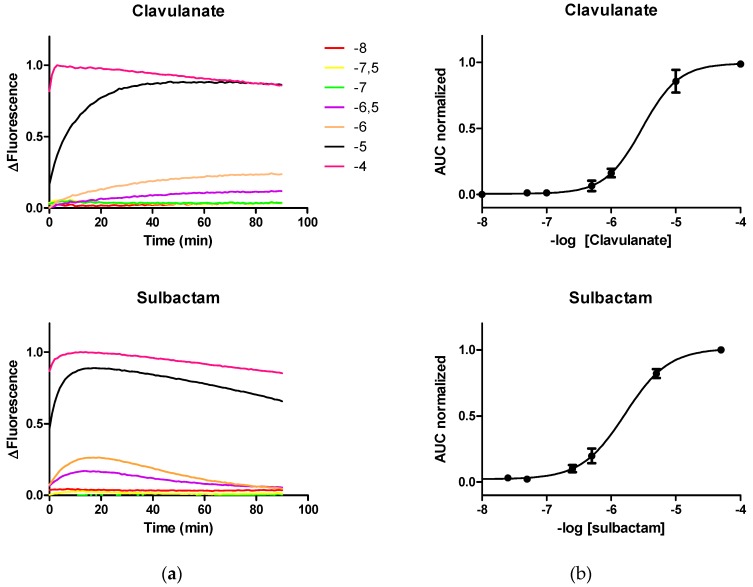
Time-course fluorescence of PenP at different concentrations of clavulanic acid (**a**) and sulbactam (**b**) Boltzmann curves, for each inhibitor, of each area under the curve normalized of fluorescence against -log concentration of antibiotic.

**Figure 3 sensors-19-01237-f003:**
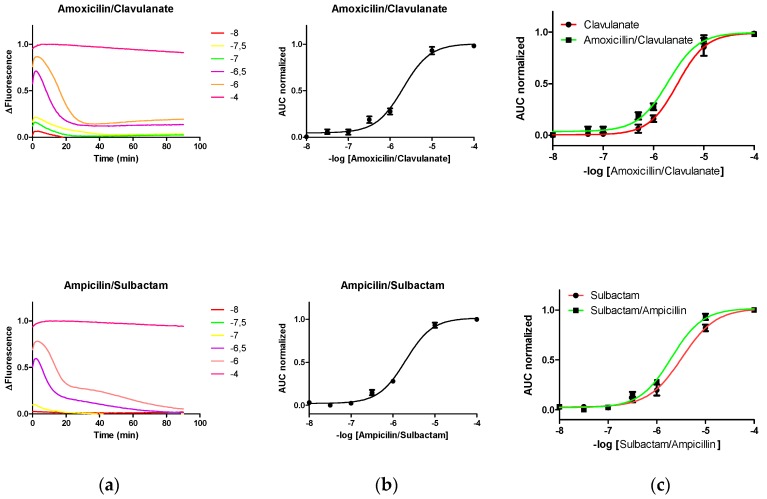
Time-course fluorescence of PenP with different concentrations of amoxicillin/clavulanic acid (**a**) and ampicillin/sulbactam (**b**) Boltzmann curves, for each composition, of area under the curve normalized of fluorescence against –log concentration of antibiotic; (**c**) Effects of LI on PenP curves at the same LA concentrations.

**Table 1 sensors-19-01237-t001:** Chemical structures of agents containing a lactam ring: Lactam antibiotics (LA) and Lactamase Inhibitors (LI).

LA(Lactam Antibiotics)	Structure	LI(Lactamase Inhibitors)	Structure
Meropenem	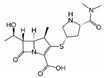	Clavulanate	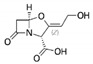
Cefazolin	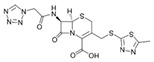	Sulbactam	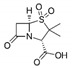
Amoxicillin	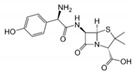		
Ampicillin	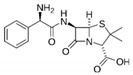		
Lactam Ring	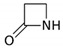

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
