# Peer review of "Effect of Lactamase Inhibitors on the Biosensor Penp during the Measurement of Lactam Antibiotics Concentration"

_sensors, 2019, doi:10.3390/s19051237_

Round 1
Reviewer 1 Report
The paper need significant improvement, if the authors really would like to publish in this journal
- line 18 and line 32 - wrong sentences - "The sensor PenP is a mutant lactamase"
- "biosensor PenP is a mutant lactamase" - biosensor it is not an enzyme...it is an anlytical tool using immobilized enzyme, in direct connection with an transducer..
- line 58 - "increase of the amount of sensor in bounded-state"... the sensor is the device... the enzyme is the biological active part of the biosensor... confusion of the terms
- line 67-68 - reagents used for the study - "acquired from local suppliers"? Need to be indicated company, purity, other info that could influence future study
- line 93 - which version of GraphPad Prism was used?
- fig 1 - a or A?
The indications are low readable, the quality of graphs are poor!!
- Fig 2 - Italian words on the graphs
- Fig 3 - Italian words on the graphs
- the discussion part is very poor - no real discussions... no comparison with other published data, novelty of the work,
- error typing - need to be verified all the text - line 184 - necesray; line 185 - Penp ....
- reference style citation in text - in the manuscript instruction - number in the brackets
- No real sample tested - to verify the real applicability and to sustain their suppositions/experiments...
Author Response
Reviewer 1; attached new version
-line 18 and line 32 - wrong sentences - "The sensor PenP is a mutant lactamase"
R: We agree and introduce a sentence “structurally derived from the mutant C166G of lactamse TEM-1”.
- line 58 - "increase of the amount of sensor in bounded-state"... the sensor is the device... the enzyme is the biological active part of the biosensor... confusion of the terms
R: We agree and eliminate unsupported sentences.
- line 67-68 - reagents used for the study - "acquired from local suppliers"? Need to be indicated company, purity, other info that could influence future study
R: We agree and introduce sources.
- line 93 - which version of GraphPad Prism was used?
R: We indicate version 5.
- fig 1 - a or A?
R: Was corrected in the graph.
The indications are low readable, the quality of graphs are poor!!
The graph were re-introduce in high quality
- Fig 2 - Italian words on the graphs
R: Versus was corrected by against.
- Fig 3 - Italian words on the graphs
R: Versus was corrected by against.
- the discussion part is very poor - no real discussions... no comparison with other published data, novelty of the work,
R: Now the Comparison as well as the novelty of the work is explicit
- error typing - need to be verified all the text - line 184 - necesray; line 185 - Penp .…
R: Was corrected.
- reference style citation in text - in the manuscript instruction - number in the brackets
The references were revised and numbering style used
- No real sample tested - to verify the real applicability and to sustain their suppositions/experiments
R: Real samples were tested in previous publication (2). In this work, we solved the effect of lactamse inhibitors on a biosensor (PenP) based on lactamase, respect to its analytical qualities, in order to can test samples with lactam antibiotics in mixtures with lactamase inhibitors which are more common in clinical procedures.

Reviewer 2 Report
- Authors did not define what was detection target chemical, for example, even in introduction part. Potential readers cannot alway find a paper's reference papers. Therefore, there should be a kind introduction to explain its theme. This paper lacks in that point. Most of all, Biosensor PrnP is not a sensor that eveyone knows already. There is no sound introduction about it.
- The scientific meanings of this paper were not significant and even it is obscure. lactam ring, antibiotics, lactamase inhibitor, etc are very confusing to understand in this paper. Authors did not pay attentions to a basic structure of paper.
- Here, the sensor seems to be a kind of enzyme from reviewer's understanding. Authors commented on "fucntional mass of sensor", which totally cannot be accepted as a scientific meanings. There should be a biochemical reaction about that. However, there is no way to find that aspects in this paper.
- There are many chemical structures like antibiotics and other inhibitors. However, no chemical structures were given.
- There are many mistakes in grammar.
Author Response
Reviewer 2; attached new version
- Authors did not define what was detection target chemical, for example, even in introduction part. Potential readers cannot alway find a paper's reference papers. Therefore, there should be a kind introduction to explain its theme. This paper lacks in that point. Most of all, Biosensor PrnP is not a sensor that eveyone knows already. There is no sound introduction about it.
R: We appreciate this feedback and we try to develop a more compressible introduction.
- The scientific meanings of this paper were not significant and even it is obscure. lactam ring, antibiotics, lactamase inhibitor, etc are very confusing to understand in this paper. Authors did not pay attentions to a basic structure of paper.
R: We agree that the functional interaction among lactam antibiotics, lactam inhibitors and a biosensor based on a mutant lactamase lead to confusion. In order to facilitate the main message we introduce the acronyms LI (for lactamse inhibitors), LA (for lactam antibiotics) and a new straightforward structure.
- Here, the sensor seems to be a kind of enzyme from reviewer's understanding. Authors commented on "functional mass of sensor", which totally cannot be accepted as a scientific meanings. There should be a biochemical reaction about that. However, there is no way to find that aspects in this paper.
R: The main idea around of the ideal behavior sensor and its relation with the concept of “functional mass available to sense” was introduced in a straightforward manner.
- There are many chemical structures like antibiotics and other inhibitors. However, no chemical structures were given.
R: Acronyms were introduced.
- There are many mistakes in grammar.
R: The whole text was revised.

Reviewer 3 Report
· The paper needs significant revisions
· The introduction do not provide sufficient background on the biosensor PenP.
· Figure 1: The quality of the figure is very bad and there is not correspondence between text, figure and legend.
· Figure 2: There are Spanish words on the graphs and there is not correspondence between text, figure and legend.
· Figure 3: There are Spanish words on the graphs
· Several error typing
· The main weak point of the paper is that the authors do not test real samples.
· Conclusion and discussion are really poor
Author Response
Reviewer 3; Attached new version
The paper needs significant revisions
R: The text and the concepts developed were revised.
· The introduction do not provide sufficient background on the biosensor PenP.
R: We appreciate this feedback and we try to develop a more compressible introduction.
· Figure 1: The quality of the figure is very bad and there is not correspondence between text, figure and legend.
The graph were re-introduce in high quality, the correspondence were double checked
· Figure 2: There are Spanish words on the graphs and there is not correspondence between text, figure and legend.
R: Versus was corrected by against.
· Figure 3: There are Spanish words on the graphs
R: Versus was corrected by against.
· Several error typing
The whole text was revised.
· The main weak point of the paper is that the authors do not test real samples.
R: Real samples were tested in previous publication (2). In this work, we solved the effect of lactamse inhibitors on a biosensor (PenP) based on lactamase, respect to its analytical qualities, in order to can test samples with lactam antibiotics in mixtures with lactamase inhibitors which are more common in clinical procedures.
· Conclusion and discussion are really poor
R: We appreciate this feedback and we try to develop a more compressible conclusion and discussion without introduce additional concepts.

Round 2
Reviewer 1 Report
there are still confusion of terms - biosensor/sensor/enzyme...
The graph quality was improved but the axes are still not in English
Real samples are still missing - the explanation of the authors is not satisfactory
Author Response
There are still confusion of terms - biosensor/sensor/enzyme...
We have tried to be more rigorous in the use of terms in the paper. For the case, we have introduced the IUPAC definition of biosensor, and based on that explicit definition that we use throughout the paper, we have developed the main idea
The graph quality was improved but the axes are still not in English
The graphics and their legends were carefully reviewed and corrected
Real samples are still missing - the explanation of the authors is not satisfactory
Currently, we are evaluating the regional (respect to injury) concentration of LA in lung tissue during acute respiratory distress syndrome (ARDS). Due to the LA composition used in the test comprise LI the validation of results using the biosensor requires previously determining the effect of LI on the analytical properties of PenP, to know the correction factors necessary to introduce in the previously developed method. In short, the real samples exist but we have reserved them to inform the results of a complete clinical study, and this study requires the characterization carried out in this work.

Reviewer 2 Report
- Authors should provide more information about PenP. For example, PenP is a lactamase, which is ** kDa, ......
- In page of 2, please put strain name of "mutant C166G", plase.
- How can readers know the 166 cysteine in PenP. Please give information, for example, PenP is a lactamase, which is *** amino acid.....
- Authors need to explain more about "time course fluorescence behavior.
- Authors should provide chemical structures of lactam antibiotics of meropenem, cefazolin, benzylpenicillin, ampicillin, ..... It will be defintely helpful, it is required.
- The graphs should have English legendary. Readers are not familar with Espanol. Check the y axis of Figure 1 and 3.
- In page of 5, LI can help to find out the activity of LA by PenP. Then the binding sites can be different from each other of LI and LA. Please discuss authors' discussions about the mechanisms of LI/LA's disparate functions. Are they competative? non-competative? or un-competative? Please refer to enzyme kinetics and discuss more.
- Authors ahould provide other beta-lactamases as a sensor for LA other than PenP.
- Discussion is not clearly described. For example, there is no comments from the results of Figure3.
- What is your LOD(limit of detection) toward lactamic antibiotics by PenP? How was the LOD changed by the introduction of lactamase inhibitors?
Author Response
- Authors should provide more information about PenP. For example, PenP is a lactamase, which is ** kDa, ......
- In page of 2, please put strain name of "mutant C166G", plase.
- How can readers know the 166 cysteine in PenP. Please give information, for example, PenP is a lactamase, which is *** amino acid.....
We agree that the absence of certain definitions led to misinterpretations and thus the lack of explicit qualities of the biosensor. Now in the introduction we have added a pair of paragraphs where we define the term biosensor and its parts, followed by the structural definition of the enzymatic part of PenP and its derived functional qualities
- Authors need to explain more about "time course fluorescence behavior.
Previously we observed that after contacting the biosensor with LA, the level of fluorescence in it changed over time (fluorescence kinetics). In this work we demonstrate that the determinations are insufficient to determine the concentration of any LA, especially those that induce a transient of fluorescence. On the other hand, in this work we showed that the integral of the PenP fluorescence for 15-90 minutes leads to a modelable curve that allows to predict the concentration of any LA.
We agree that the above is not entirely obvious in this work, but we have privileged the exclusion of contents previously disclosed. Notwithstanding this, we have tried to improve the description of kinetics of the fluorescence and fluorescent profile in the introduction, in materials and methods, and in the discussion
- Authors should provide chemical structures of lactam antibiotics of meropenem, cefazolin, benzylpenicillin, ampicillin, ..... It will be defintely helpful, it is required.
New table 1 has been introduced
- The graphs should have English legendary. Readers are not familar with Espanol. Check the y axis of Figure 1 and 3.
The graphics and their legends were carefully reviewed and corrected
- In page of 5, LI can help to find out the activity of LA by PenP. Then the binding sites can be different from each other of LI and LA. Please discuss authors' discussions about the mechanisms of LI/LA's disparate functions. Are they competative? non-competative? or un-competative? Please refer to enzyme kinetics and discuss more.
Because the interaction site of LI and LA is the same, a competitive type mechanism was expected. When considering that LI have an irreversible interaction, a progressive reaction was also expected. Because the graphs of LA with or without LI are extraordinarily similar is that the propositions of competition and progression are discarded, and we can only speculate (given the lack of mechanistic tests) the presence of steric hindrance. The latter now explicit in the text.
- Authors ahould provide other beta-lactamases as a sensor for LA other than PenP.
Dr. Wong is working on a lactamase mutant type C (with the substitution V211C) instead of the TEM-1 type A (with the substitution G166). But based on the preliminary results, this is a new job by itself.
- Discussion is not clearly described. For example, there is no comments from the results of Figure3.
A new discussion incorporating the new figure 3, its results and explicit quantification has been incorporated.
- What is your LOD(limit of detection) toward lactamic antibiotics by PenP? How was the LOD changed by the introduction of lactamase inhibitors?
A new discussion incorporating the new figure 3, its results and explicit quantification has been incorporated.

Reviewer 3 Report
· Introduction on PenP is still poor. Provide more chemical information about the biosensor PenP and describing also what is a biosensor in general.
· The graphs still have Spanish legend (x axis: “Tiempo”).
· Provide chemical structures of lactam antibiotics that you mentioned, please.
· In the Figures 2 and 3 put the letters A and B, please.
· Discussion is still not described with clarity and experiments using real samples are still missing.
Author Response
Introduction on PenP is still poor. Provide more chemical information about the biosensor PenP and describing also what is a biosensor in general.
We have tried to improve and enrich the information provided in the introduction in order to make the main objective of the study more evident. In this occasion, we have introduced paragraphs that include abstract, structural and functional definitions. We have tried to explain better the mechanisms that support this study. In this sense, a new table containing structural information has been introduced.
· The graphs still have Spanish legend (x axis: “Tiempo”).
The graphics and their legends were carefully reviewed and corrected
· Provide chemical structures of lactam antibiotics that you mentioned, please.
New table 1 has been introduced
· In the Figures 2 and 3 put the letters A and B, please.
The graphics and their legends were carefully reviewed and corrected
· Discussion is still not described with clarity and experiments using real samples are still missing.
A new discussion incorporating the new figure 3, its results and explicit quantification has been incorporated.
Currently, we are evaluating the regional (respect to injury) concentration of LA in lung tissue during acute respiratory distress syndrome (ARDS). Due to the LA composition used in the test comprise LI the validation of results using the biosensor requires previously determining the effect of LI on the analytical properties of PenP, to know the correction factors necessary to introduce in the previously developed method. In short, the real samples exist but we have reserved them to inform the results of a complete clinical study, and this study requires the characterization carried out in this work.

Round 3
Reviewer 1 Report
there are still mistache associated
- each molecule of Biosensor - page 5, line 131
Low number of reference used...
Author Response
We sincerely appreciate the suggestions that have allowed us to improve this work, and the recognition of the effort we have made in that regard.
there are still mistache associated
We have done a thorough review of the grammar in the paper, and we have made the necessary corrections.
- each molecule of Biosensor - page 5, line 131
The discussion of the effect of LI on the biosensor, based on the quality of the molar ratio between these agents, was rewritten.
Low number of reference used...
New references that facilitate access to prior art have been incorporated

Reviewer 2 Report
- Introduction part was improved to give readers kinder introduction.
- The first part of introduction about "definition of IUPAC" is not necessary.
- There are still lots of grammatic mistakes in English writings.
Author Response
- Introduction part was improved to give readers kinder introduction.
We sincerely appreciate the suggestions that have allowed us to improve this work, and the recognition of the effort we have made in that regard.
- The first part of introduction about "definition of IUPAC" is not necessary.
We think in a similar way when considering the target audience to which this journal is directed. However, another reviewer has insisted on the use of stringent academic definitions that allow to differentiate a modified mutant enzyme from a biosensor. Therefore, we have eliminated the IUPAC reference but will insist respectfully on maintaining the definition of terms.
- There are still lots of grammatic mistakes in English writings.
We have done a thorough review of the grammar in the paper, and we have made the necessary corrections.

Reviewer 3 Report
I appreciated the work of authors to improve it.
Just the English should be improved a little bit more.
Author Response
I appreciated the work of authors to improve it.
We sincerely appreciate the suggestions that have allowed us to improve this work, and the recognition of the effort we have made in that regard.
Just the English should be improved a little bit more.
We have done a thorough review of the grammar in the paper, and we have made the necessary corrections.
